# Effects of Fiber Loading on Mechanical Properties of Kenaf Nanocellulose Reinforced Nanohybrid Dental Composite Made of Rice Husk Silica

**DOI:** 10.3390/jfb14040184

**Published:** 2023-03-27

**Authors:** Su Bing Sheng, Rabihah Alawi, Yanti Johari, Nor Aidaniza Abdul Muttlib, Mohd Hazwan Hussin, Dasmawati Mohamad, Mohmed Isaqali Karobari

**Affiliations:** 1School of Dental Sciences, Universiti Sains Malaysia, Health Campus, Kubang Kerian, Kota Bharu 16150, Kelantan, Malaysia; 2School of Chemical Sciences, Universiti Sains Malaysia, Minden 11800, Penang, Malaysia; 3Conservative Dentistry Unit, School of Dental Sciences, Universiti Sains Malaysia, Health Campus, Kubang Kerian, Kota Bharu 16150, Kelantan, Malaysia; 4Department of Conservative Dentistry & Endodontics, Saveetha Dental College & Hospitals, Saveetha Institute of Medical and Technical Sciences University, Chennai 600077, Tamil Nadu, India

**Keywords:** nanocellulose, kenaf, dental composites, fiber-reinforced composites, flexural strength, compressive strength

## Abstract

The innovation of nanocellulose as reinforcement filler in composites has been a topic of interest in the development of new biomaterials. The objective of this study was to investigate the mechanical properties of a nanohybrid dental composite made of rice husk silica and loaded with different percentages of kenaf nanocellulose. Kenaf cellulose nanocrystals (CNC) were isolated and characterized using a transmission electron microscope (TEM) (Libra 120, Carl Zeiss, Germany). The experimental composite was fabricated with fiber loadings of 1 wt%, 2 wt%, 3 wt%, 4 wt%, and 6 wt% silane-treated kenaf CNC, and subjected to a flexural and compressive strength test (*n* = 7) using an Instron Universal Testing Machine (Shimadzu, Kyoto, Japan), followed by a scanning electron microscopic assessment of the flexural specimen’s fracture surface using a scanning electron microscope (SEM) (FEI Quanta FEG 450, Hillsborough, OR, USA). Commercial composites Filtek Z350XT (3M ESPE, St. Paul, MN, USA), Neofil (Kerr Corporation, Orange, CA, USA) and Ever-X Posterior (GC Corporation, Tokyo, Japan) were used as a comparison. The average diameter of kenaf CNC under TEM was 6 nm. For flexural and compressive strength tests, one-way ANOVA showed a statistically significant difference (*p* < 0.05) between all groups. Compared to the control group (0 wt%), the incorporation of kenaf CNC (1 wt%) into rice husk silica nanohybrid dental composite showed a slight improvement in mechanical properties and modes of reinforcement, which was reflected in SEM images of the fracture surface. The optimum dental composite reinforcement made of rice husk was 1 wt% kenaf CNC. Excessive fiber loading results in a decline in mechanical properties. CNC derived from natural sources may be a viable alternative as a reinforcement co-filler at low concentrations.

## 1. Introduction

Physical and mechanical properties are major determinants of composite resin restorations (CRR) [1]. Fillers are mainly responsible for the physical and mechanical properties of CRR, and fillers have been extensively studied following the development of composite resin materials. The incorporation of fibers into dental composites enhances fracture toughness and flexural strength, as stress is transferred from a weaker resin matrix to stronger fine fibers, hence acting as crack stoppers [2,3,4]. The high strength of E-glass fibers is responsible for their wide usage in dental composites. However, irritation to the body and suspected carcinogenicity are among the health concerns and limitations of using E-glasses [5,6,7].

Researchers have attempted to derive reinforcement fillers from natural sources as substitute synthetic materials and move towards a green sustainable future. Natural fibers are of great abundance on earth and are considered viable alternatives to mineral-based reinforcing fibers. Key benefits of lignocellulosic fibers over inorganic fillers include being renewable, their wide variety, low energy consumption, non-food agricultural-based economy, non-abrasive nature, cost-effectiveness, high specific strength and modulus, and low density [8].

Further insight into lignocellulosic fibers has led to the discovery of nanocellulose and unveiled a wide range of opportunities in various fields. Nanocellulose is advantageous in comparison to its micro-sized counterparts because of its larger surface area, which facilitates a larger contact surface between the matrix and fiber for better adhesion [9,10]. The three main types of nanocellulose are cellulose nanofibrils (CNF), nanocrystals (CNC), and bacterial cellulose (BC) [11,12]. The high crystalline chains present in CNCs contribute to their high structural stiffness and strength in comparison to other nano-sized forms [13,14,15]. The incorporation of CNCs into glass ionomer cement and flowable composites reflects a considerable increase in mechanical properties [9,16].

Previous research revealed that kenaf fiber is promising as a reinforcing fiber in thermoplastic composites, which is linked to its remarkably high aspect ratio and toughness relative to other fibers [17,18,19]. The specific stiffness is comparable to the glass fibers, with twice or thrice the cost reduction compared to glass fibers [19,20]. Empirically, a significantly higher flexural strength was demonstrated by kenaf-reinforced polymethyl methacrylate (PMMA) compared to the conventional material [21].

In 2016, rice husk was successfully utilized as a source of nanohybrid silica particles and used as fillers in dental composite [22]. However, it was found that rice husk silica nanohybrid dental composite exhibited inferior flexural and compressive strength, as well as Vicker’s hardness, compared to the commercial product, which the author attributed to the difference in filler loading, as well as the incorporation of co-fillers in commercial products [22]. Despite not being comparable to commercial products, rice husk-derived nanohybrid silica composite resin could be further improved and developed [23]. In this study, an attempt was made to address the mechanical deficiency of nanohybrid rice husk silica composite resin by incorporating kenaf CNCs as co-fillers to fabricate fiber-reinforced composite, and concurrently widen the application of nanocellulose in the field of dentistry.

In summary, kenaf CNCs were extracted from their raw fiber source, characterized, and subsequently silane-treated, before being added to a nanohybrid rice husk composite resin mixture in a different fiber loading. The experimental composite resin was tested for flexural and mechanical strength, where the fracture surface of the flexural surface was analyzed for fracture features, the quantity of fiber, integrity, and homogeneity along the surface between fiber and matrix. The null hypothesis of this study (H0) was that increasing the kenaf CNCs will not improve the flexural and compressive strength of rice husk silica nanohybrid dental composite.

## 2. Materials and Methods

### 2.1. Extraction of CNC from Kenaf

The CNCs were prepared from raw Grade A kenaf fibers (SKM2—Bio Grade A) purchased from National Kenaf and Tobacco Board, Kota Bharu, Kelantan, Malaysia. Figure 1 shows the extraction of kenaf CNC from raw kenaf fibers. Running filtered water was used to wash and rinse the samples and dried them at 50 °C for 24 h. Fibers were then milled and separated through a 100-mesh sieve. Fibers above the sieve size of 63 µm, and below the sieve size of 150 µm were excluded, limiting the size between 63 and 150 µm [24].

In terms of dewaxing, 10 g kenaf fibers were filled into a cotton thimble and immersed in a Soxhlet reflux apparatus attached to a round-bottom flask filled with 200 mL ethanol and 100 mL toluene (ethanol: toluene *v*/*v* 2:1). The Soxhlet reflux system was heated uniformly via round bottom flask immersed in paraffin oil bath for five hours to remove waxes and extractives [25]. Fiber suspension was filtered through a filter paper layered on a Büchner funnel, which was attached to a Büchner flask connected to running water via rubber tubing to create a partial vacuum to speed up filtration. Subsequently, distilled water was used to wash the fibers through the Büchner filtration system. Fibers were removed and placed in an oven at 50 °C to dry overnight.

Regarding alkali treatment, 5 g of extractive-free fibers were immersed in 200 mL 4% NaOH solution for two hours at 80 °C under constant mechanical stirring using an overhead stirrer and filtered until a neutral pH was achieved, before drying in an oven at 50 °C overnight. Meanwhile, bleaching treatment was performed by adding 1 g of alkali-treated fibers to a mixture containing distilled water, acetic acid, and 25% sodium chlorite (NaClO) at 125, 1, and 3 mL, respectively. Mechanical stirring was also applied for the bleaching treatment for two hours under reflux at 70 °C before filtering and oven-drying at 50 °C overnight.

The CNCs from kenaf cellulose fibers were extracted via acid hydrolysis as described by Hussin et al. [26]. Specifically, treated kenaf fibers were hydrolyzed in 25 mL of 50% sulfuric acid for 30 min. Distilled water was used to dilute the precipitate-containing CNC four times by centrifuging for 10 min at 7900 rpm using a centrifuge machine (Eppendorf, Hamburg, Germany). Dialysis of the precipitate was performed by filling the precipitate into a dialysis tubing cellulose membrane, which was tied and immersed in tap water until a neutral pH of 7 was achieved. The pH was checked using Whatman pH indicator papers. CNCs obtained were freeze-fried for 24 h. A transmission electron microscopy (Libra 120, Carl Zeiss, Oberkochen, Germany) was then used to visualize the extracted kenaf CNC. The summary of the extraction process of kenaf CNC from raw kenaf fiber is illustrated in Figure 1.

### 2.2. Treatment of CNC with Silane

The extracted CNC was treated with silane using the method described by Noushad et al. [22]. Nanocellulose that was already soaked in ethanol and water solution (ratio of 70:30 and 20:1 liquid/solid ratio) was added with about 1 wt% of γ-methacryloxypropyltrimethoxysilane (MPS), based on the Brunauer–Emmett–Teller (BET) surface area analysis. The extracted kenaf CNC showed a BET surface area of 18.894 m^2^/g, with average pore width of 1.32 nm. According to Hussin et al. [26], the CNC was classified as microporous as the pore size was less than 2 nm.

Treatment with silane was conducted for 4 h at 37 °C and stirred mechanically. Excess γ-MPS was washed off with acetone, whereas silane-treated CNC was oven-dried at 80 °C for 24 h. The untreated and silane-treated kenaf CNC were sent for Fourier-transformed infrared spectroscopy (FTIR) using an FTIR spectrometer (PerkinElmer Spectrum 100, Shelton, WA, USA).

### 2.3. Composite Resin Fabrication

Silanized rice husk silica was purchased from USAINS, School of Dental Sciences, Universiti Sains Malaysia. Rice husk silica particles are spherical, with a diameter ranging from 48 to 534 nm [22]. Bisphenol A-glycidyl methacrylate (Bis-GMA), triethylene glycol dimethacrylate (TEGDMA), and Dimethylamino ethyl methacrylate (DMAEMA) were initially mixed before adding camphorquinone. The compositions of monomers were proportioned in weight percentage (Table 1). The mixture was mixed homogeneously. Silane-treated kenaf CNC was then added and mixed manually, followed by silanized rice husk silica, to produce a homogenous paste based on the experimental groups (see Table 1). The percentage of CNC used in this study ranged from 1% to 6%. The comparisons included commercial composite Neofil (Kerr Corporation, Orange, CA, USA), Ever-X posterior (GC Corporation, Tokyo, Japan), and Filtek Z350XT (3M ESPE, St. Paul, MN, USA).

### 2.4. Mechanical Strength Testing

Experimental fiber-reinforced composites were prepared for seven flexural strength specimens with a slotted stainless-steel mold per ISO 4049:2019 standard of (25 × 2 × 2 mm) and light cured with a light cure unit at 1000 mW/cm^2^ (Elipar Deep Cure, 3M, St. Paul, MN, USA). Each increment was light cured for 40 s with an overlapping cure regimen based on ISO 4049. The samples were preserved at room temperature for 24 h in distilled water post-polymerization. The samples were tested using an Instron Universal Testing Machine manufactured by Shimadzu, Japan, with a loading force of 5 kN and a crosshead speed of 0.75 mm min^−1^.

Concerning compressive strength samples, the specification used in preparing the experimental composites (*n* = 7) was the ASTMD 695-08 of 4 mm diameter × 6 mm height. This specification was achieved by using a stainless-steel mould with incremental light curing protocol with a light cure unit of 1000 mW/cm^2^ (Elipar Deepcure L, 3M, St. Paul, MN, USA). Each increment was light cured for 40 s. Likewise, the samples were stored and tested as mentioned previously, excluding 1 mm min^−1^ crosshead speed.

### 2.5. Fractured Samples’ Surface Analysis

The fracture surface morphology was assessed by arbitrarily selecting fractured flexural samples from each group. Chosen samples were shortened to avoid contacting the SEM microscope. The method is shown in Figure 2. Metal stubs were used as fixation medium for the samples, which were subsequently sputtered under a vacuum in platinum (1 cycle of 120 s). The fractured surface was then analyzed using a scanning electron microscope (FEI Quanta FEG 450, Hillsborough, OR, USA).

### 2.6. Data Analysis

All the gathered data were analyzed in the 26th version of SPSS (IBM Corp., Armonk, NY, USA). Compressive strength and flexural strength were analyzed by comparing their mean values using ANOVA. Post-hoc analysis was conducted using Dunnett’s T3 tests. Parameter estimations were estimated at 95% confidence intervals and statistically significant differences were set at a *p*-value < 0.05.

## 3. Results and Discussion

### 3.1. Kenaf CNC Characterization Using Transmission Electron Microscope (TEM)

Prior to assessment under TEM, the kenaf CNC was sent for scanning electron microscopy (SEM). The nano size of the CNC was not measurable under SEM due to agglomeration. The kenaf CNC was rod-like and presented a network structure similar to a web following TEM analysis. Overall, the kenaf CNC displayed a diameter that ranged between 5.38 nm and 8.09 nm, with a mean of 6.31 nm (Figure 3). This appearance of kenaf CNC aligns with the findings of Nuruddin et al. [27], in which a mean diameter of 6 nm was documented after characterization. The CNC lengths could not be measured under TEM following undiscernible ends and overlapping samples of elongated curved nanofibers, in line with the Nuruddin et al. [27] report. This finding stems from the agglomeration owing to strong inter-fiber hydrogen bonding. Nevertheless, CNC was successfully isolated from kenaf cellulose for utilization in various applications.

Dewaxing and alkaline treatment increase the aspect ratio by removing impurities, depolymerizing cellulose, exposing short-length crystallites, and reducing fiber diameter [18,28,29,30,31]. Meanwhile, increased tensile strength was reported by the removal of hemicellulose and lignin [32]. Lignin negatively affected the fiber-matrix interphase [33]. Based on Husnil et al. [34], some cellulose parts may be degraded by harsh bleaching conditions, thus eliciting a lower degree of polymerization.

Amorphous regions are selectively degraded during acid hydrolysis, while highly crystalline structures are preserved in the process [12,35]. Higher crystallinity was observed upon performing acid hydrolysis for 55 min compared to 45 min Sulaiman et al. [36]. Additionally, hydrolysis duration can be used to compensate for the reactions conducted at a lower temperature and lower acid concentration [37]. Nanocellulose characterized with a higher crystallinity was synthesized upon using lower acid concentration for longer time intervals Wulandari et al. [38]. The differences in sources and preparation methods contributed to the variability of nanocellulose structure and performance [39,40].

Bai et al. [41] reported that smaller precipitated CNC whiskers were produced when the centrifugal force increases as far as the centrifuge time is constant. Centrifugal fractionation is an efficient and simple technique used in fractionating a homogenous nanocellulose, in which the molecular size is uniformly distributed [42]. The present study employed 50% sulfuric acid for acid hydrolysis. The procedure was conducted for 30 min at room temperature. The homogenized size of CNC was isolated after centrifuging the sample four times at 7900 rpm for 10 min.

In comparison to the fiber, the critical length (L_c_) could be 40–50 times larger in terms of diameter [43,44]. Short fibers are more likely to elicit brittleness in composite due to having more fiber ends, thereby concentrating the stress at both ends and resulting in fracture [45]. In contrast, since long fibers are easily broken during the process and become interwoven, excessive fiber length is considered unfavorable [45]. Long fibers are also susceptible to tangling during mixing, resulting in poor fiber dispersion and reduced reinforcement efficiency [46]. Thus, proper composite reinforcement occurs when stress transfer, fiber packing, and orientation, are efficient, as well as when a fiber with an optimum length of fiber is used [45].

### 3.2. Chemical Interaction Analysis between the Silane and Kenaf CNC Using FTIR

FTIR close-up spectra of raw kenaf, kenaf cellulose, kenaf CNC, and MPS-CNC (Figure 4) show the emergence of peak 1719 cm^−1^, corresponding to the C=O group of γ-MPS [47]. A tiny peak at 785 cm^−1^ was detected and attributed to Si-CH bending [48].

FTIR of silane-treated CNC (γ-MPS-CNC) shows the presence of carbonyl stretching, which is characteristic of the -COO- group of γ-MPS [47]. γ-MPS adsorption occurs through acid-base interactions between its ester group and cellulosic OH groups, which shows a peak in 1720 cm^−1^ for the free carbonyl group [49]. The tiny peak at 785 cm^−1^ is close to a theoretical value of 793 cm^−1^ for Si-CH bending suggested by Rangel and Leal-García [48]. Strong characteristic peaks of C-O and C-O-C stretch in the 1200 to 1000 cm^−1^ region, which obstructs peak assignments for Si-O-Si and Si-O-cellulose bonds [47], suggesting that γ-MPS was adsorbed onto CNC.

### 3.3. Mechanical Strength of Composite Resin Reinforced with Silane-Treated CNC

As shown in Table 2, Ever-X Posterior composite (GC Corporation, Tokyo, Japan) demonstrated the highest flexural strength of 121.46 ± 4.49 MPa, whereas group K6 (6 wt% CNC) recorded the lowest mean flexural strength (49.24 ± 5.29) MPa. For the experimental composites, incorporating 1 wt% CNC led to an increase in flexural strength by 4.8% from 66.24 ± 7.22 to 69.46 ± 5.01 MPa (*p* > 0.05). Increasing the fiber loading from 2 wt% onwards also led to a significant reduction in flexural strength. The composite’s flexural strength was significantly reduced by the 6 wt% CNC.

Table 2 shows that overall, the flexural strength differed significantly between the groups (*p* < 0.05). Specifically, Group K1 (1 wt% CNC) recorded significantly higher mean flexural strength relative to group K3 (3 wt% CNC). Flexural strength was significantly higher in Groups K1 (1 wt% CNC) and K2 (2 wt% CNC) relative to group K6 (6 wt% CNC). Nonetheless, the commercial groups (C1, C2 and C3) recorded significantly higher mean flexural strength relative to the experimental groups (K1 to K6). The Ever-X Posterior composite (GC Corporation, Tokyo, Japan) and Group K6 (6 wt% CNC) recorded the highest and lowest mean compressive strength (244.15 ± 38.84 vs. 142.87 ± 9.01 MPa), respectively (Table 3). The composite increased by 17% (148.16 ± 26.63 to 173.51 ± 15.77 MPa) following the incorporation of CNC at 1 wt%, which is consistent with 3M Filtek composite (3M ESPE, St Paul, MN, USA). Nevertheless, the increment was not statistically significant. A reduction in the compressive strength was recorded using a 6 wt% CNC fiber loading. As depicted in Table 3, Group K6 (6 wt% CNC) recorded significantly lower mean compressive strength than Neofil (Kerr Corporation, Orange, CA, USA). Higher compressive strength was exhibited (*p* < 0.05) by the EverX Posterior (GC Corporation, Tokyo, Japan) relative to other groups, excluding Neofil (Kerr Corporation, Orange CA, USA). Mechanical properties are multifactorial, ranging from types of resin matrix to the strength of interfacial phase, degree of conversion of the matrix, types, loading, sizes, shapes, and orientation of fillers.

Given the aforementioned points, several factors were controlled in this study. For example, the same matrix and ratio (BisGMA/TEGDMA 60:40) were considered for the degree of conversion and matrix type. Meanwhile, consistent 40 s/increment in light curing protocol and overlapping cure regimen was ensured by the researcher based on ISO 4049 and a similar matrix: filler ratio. The samples were also immersed in water for one day before testing as per ISO 4049 to reduce the contraction stress. This procedure enables the water to diffuse into the material, thus counteracting the contraction stress as the volume of water increases [50]. The main issues observed in using lignocellulosic fibers to reinforce composites are poor bonding and water sorption [21,51,52]. Several episodes of chemical treatments were employed in this study to minimize this water sorption and bonding issues.

The mean flexural strength and compressive strength of 66 MPa and 148 MPa respectively were recorded for the control samples of rice husk nanohybrid silica. A flexural strength of 69 MPa and compressive strength of 173 MPa was observed in samples reinforced with 1 wt% silane-treated kenaf CNC. Resultantly, for restorations encompassing occlusive surfaces, a flexural strength of ≥80 MPa was required by ISO 4049:2019. Meanwhile, a lower flexural strength (≥50 MPa) was needed for other restorations [53]. Although the synthesized composites may differ compared to commercial products, the former attained more than 50 MPa, thus fulfilling ISO 4049:2019, as per the flexural strength for restorations apart from those involving occlusal surfaces.

The percolation theory has been employed to elucidate the process in which nanocellulose reinforces composites [54], which is used in material science when describing the transitional behavior of mechanical and electrical properties in composites [55]. The interconnection of the CNCs network within the polymetric matrix leads to the formation of a rigid supporting architecture and reinforcement. This reinforcement makes use of hydrogen bonding between CNCs to enable stress transfer [9,56]. Finite and infinite clusters were reported as the two types of filler clusters influencing mechanical properties by Noël et al. [55]. While the strongest reinforcing effect is elicited by the infinite cluster spanning across the composite, the filler fraction required to yield the first percolating path through the polymer matrix is the percolation threshold or critical volume fraction [55]. In response to the initial application of CNC for composite reinforcement, the distribution in the CNC‘s orientation, the aspect ratio, and the critical volume fraction are the main determinants of the critical volume fraction [57].

According to Onsager [58], an orientational disorder–order phase transition occurs in CNCs by attaining an orientationally ordered anisotropic phase from a disordered isotropic phase. The attainment of a critical volume fraction marks the onset of this phase transition [59]. Azizi Samir et al. [8] summarized the factors affecting phase transition, namely, the surface charge of CNCs [59,60], their geometrical axial ratio [58], and length polydispersity [61]. The interaction between the matrix and filler is important, but filler–filler interactions also affect the reinforcing capability of the CNC [8,62].

The null hypothesis of this study was accepted. Both compressive and flexural strengths depicted a decreasing trend, particularly for the kenaf with 2 wt% and 3 wt% CNC loading, respectively. This may arise from the increasing agglomeration of kenaf CNC, which induced poor fiber dispersion. As a result, voids are formed in the structure of the composite resin, thus initiating the formation of cracks. The use of a manual method in mixing the fibers into the composite resin may also contribute to poor dispersion. Microstructures characterized by weak interaction are produced by agglomerated CNCs, thereby reducing load-bearing capacity and concurrently forming voids in which clumps cannot be penetrated by polymer matrix [63]. Zarina and Ahmad [64] demonstrated that CNC agglomeration diminishes the contact at the filler matrix interface, leading to poor interfacial stress.

Each fiber ends acts as stress concentration at a low fiber weight fraction, leading to sites of weakness disintegrating when a low loading is applied [65,66]. Nonetheless, mechanical strength reduction may be expected due to fiber-fiber interaction, dispersion problems, and void contents, particularly when fiber weight fraction is excessive [66]. Purification and sonication are treatments that promote the dispersion of the CNCs of hydrolyzed celluloses [67].

Inadequate polymerization has been postulated as the factor responsible for the deterioration in mechanical properties of high fiber loading in the composite owing to the obstruction of light transmission through the composite resin by nanocellulose. The refractive index (RI) of TEGDMA and BisGMA is 1.46 and 1.54, respectively [68]. Wang [69] found that the RI of CNC in their samples ranged from 1.473 to 1.488, and the values were influenced by surface modifications. Nevertheless, their results were not specifically related to kenaf. It is recommended that the diameters of nanoparticles should be less than 40 nm to synthesize transparent and high RI composite materials. Such designs will assist in maintaining the intensity of the light transmitted and prevent Rayleigh scattering [70,71]. Light scattering may also be enhanced due to the percolation effects associated with nanofillers of small diameters, given their negative impact on matrix transparency. Hence, researchers posited that the optical and mechanical properties represented dual antinomic features [70,72].

Natural fibers have been successfully incorporated into reinforcing composites in dentistry. Thamara et al. [73] incorporated bagasse fiber into a bulk-fill composite resin, which led to a significant increase in the compressive strength from 348 MPa to 353 MPa, but still less than the values observed in the composite with synthetic E-glass fiber (364 MPa) [74].

The inferior mechanical properties in K6 (6 wt% CNC) demonstrated the consequence of an excessive increase in fiber loading. The various types, sizes, and densities of fillers used in commercial composites may contribute to the significant difference in mechanical strength relative to experimental composites. Notwithstanding, more efforts are needed to optimize the filler loading, degree of conversion, resin matrix/photoinitiator ratio, and depth of cure of rice husk silica composite resin and improve their overall mechanical properties. In addition, the effects of different densities between rice husk silica, kenaf CNC and glass fillers used in the commercial dental composite on the mechanical strength shall be further explored.

### 3.4. Scanning Electron Microscope (SEM) Analyses of the Fractured Surface

Figure 5a–f depicts the K0 to K6 samples as observed in the SEM (SEM images at 5000× magnifications). Essentially, bundle and single-stranded CNCs were observed on the cross-sectional area of fractured samples. It can be seen that CNCs dispersed and embedded in the matrix, as seen in Figure 5b–f. The circular shape of CNC can be seen in a cross-section view. Some porosities were noted inside the CNCs (Figure 5f). K0 sample depicted no nanocellulose and the surface appears regular without porosities (Figure 5a). The surface topology of the sample appeared irregular upon nanocellulose incorporation, especially in sample K1 (1 wt% CNC). Overall, a small gap was observed circumferentially between the matrix and the CNC (Figure 5). Figure 5b–f depicts the presence of specks of particulates on the surface of the pulled-out CNCs. More CNCs were seen with increasing CNCs loading from sample K1 to K6, concurrent with evidence of the increasing occurrence of porosities. Crack propagation stopped at CNC in K3 (3 wt% CNC) and K6 (6 wt% CNC) samples (see Figure 5d,f). Deflection of crack propagation was observed in the K4 (4 wt% CNC) sample (see Figure 5e).

Failure types of discontinuous short fibers include matrix failure, fiber fracture, and fiber pull-out or debonding [43,47], which are influenced by the aspect ratio of fiber, length of the fiber, fiber volume fraction, and interfacial fracture energy. Matrix failure is expected to occur initially, following the transfer of stress to reinforcement fibers, resulting in fiber pull-out and failure. All the samples displayed pull-out CNC failure, thus suggesting insufficient CNC-matrix adhesion. Nonetheless, some parts of interfacial adhesion may arise, as the surface appeared rough with existing particulates at the surface of the CNCs pulled out. Short pull-out CNCs may reflect the possibility of adhesion failure before debonding. Inadequate adhesion may also be indicated in the microscopic gap located around the CNCs circumferentially. Notwithstanding this, it was still evident that the CNC employed stress transfer in reinforcing the composite. The fracture surface was regular in the K0 sample (0 wt% CNC), due to the absence of CNC reinforcement. A brittle fracture is indicated by the appearance of a smooth fracture surface, whereas a rougher crack surface denotes a deflection mechanism, culminating in higher crack propagation resistance [75]. The surface in all CNC-reinforced samples appeared more irregular, thus suggesting the deflection of cracks by CNCs into another plane. In K3 (3 wt% CNC), the CNC acted as a crack stopper, while those in the K4 sample (4 wt% CNC) appeared to deflect the direction of the crack. More voids were observed with increased CNC loading from the K2 sample (2 wt% CNC) onwards (Figure 5c–f).

In this study, no investigation was performed for the mechanical strength of individual CNC, particularly in terms of CNC processing parameters. A percolating network formation is expected to elicit CNCs with stronger mechanical strength, in which factors such as CNC-matrix interaction, critical concentration, and filler-filler interaction determine the critical concentration. Nevertheless, no percolating network was synthesized in this study relative to the report by Silva, Pereira [9].

### 3.5. Limitations of the Study and Future Recommendations

The limitations of this study largely revolve around CNC aggregation. With an increasing aspect ratio, the risk of agglomeration is expected to increase. Such clumping would reduce the efficiency of chemical processes, particularly surface modification, ultimately affecting the adhesion of CNC to the resin matrix. CNC agglomeration with undiscernible ends disables length measurement under TEM. During the mixing of composite resins, inevitable variations in manual hand-mixing motion may affect the resultant mechanical performance of the composite resin. The limitation of this study includes the uncertain microstructure of kenaf CNC, as well as the unquantified density of kenaf CNCs. Furthermore, the degree of conversion was not verified and may affect the resultant mechanical strength.

Fiber loading of less than 6 wt% and at closer intervals may be considered in future studies to elucidate their impacts on mechanical properties. Future research may also focus on optimizing kenaf interfacial adhesion and processing. In order to prevent fiber tangling, the fiber size could be optimized further to generate shorter rod-like CNCs. The best mechanical properties could be derived by optimizing the parameters of kenaf composite fabrication and fiber processing formulae. Fiber ultrasonication can be explored to enhance the dispersion of kenaf composites. Another method of silanization protocol of CNCs should be explored by using another type of silane treatment, such as (3-Aminopropyl) triethoxysilane (APTES). Further testing on biocompatibility, antimicrobial, and ageing characteristics of mechanical properties is required shortly. Nanocellulose could be better employed in reinforcing co-fillers in dental composites by eluding the reinforcement mechanisms of CNCs.

## 4. Conclusions

In conclusion, kenaf CNC remains a viable reinforcement co-filler in dental composites given its potential to induce significant enhancements in low-volume fractions. This was reflected in this study as the most optimal mechanical performance with 1 wt% fiber loading. Nevertheless, future studies should consider exploring the potential of kenaf CNCs as reinforcement fillers by addressing the challenges related to the dispersion of CNCs and fiber-matrix adhesion in the polymer matrix.

## Figures and Tables

**Figure 1 jfb-14-00184-f001:**
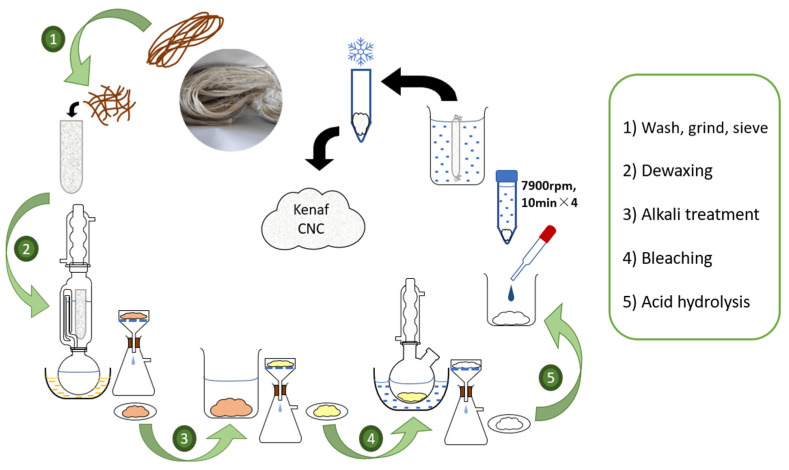
The extraction process of kenaf CNC from raw kenaf fiber.

**Figure 2 jfb-14-00184-f002:**
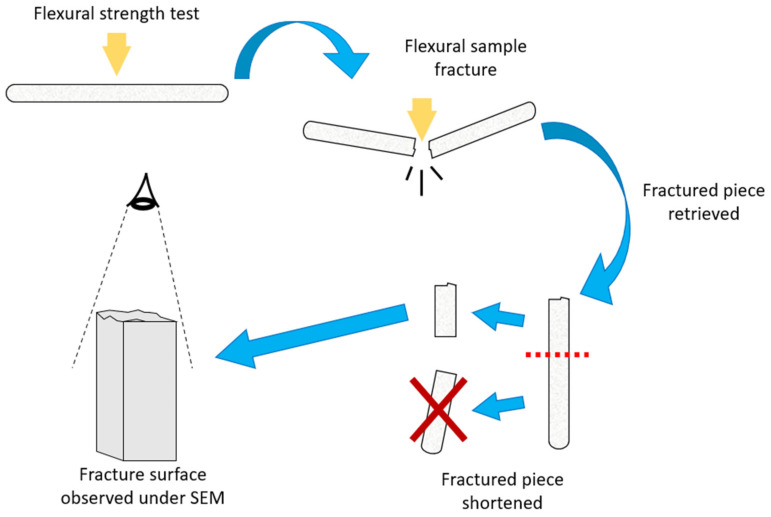
Method for fracture surface analysis of flexural samples.

**Figure 3 jfb-14-00184-f003:**
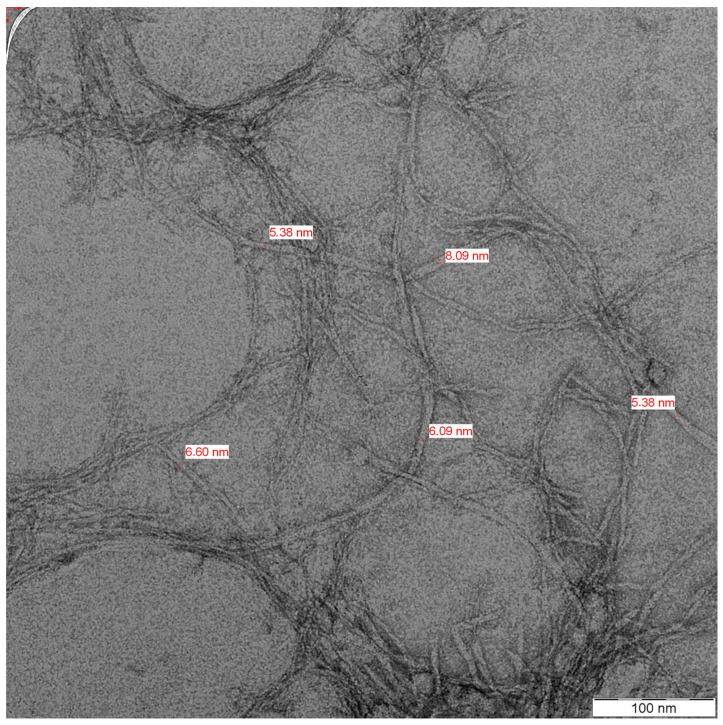
TEM image of kenaf CNC.

**Figure 4 jfb-14-00184-f004:**
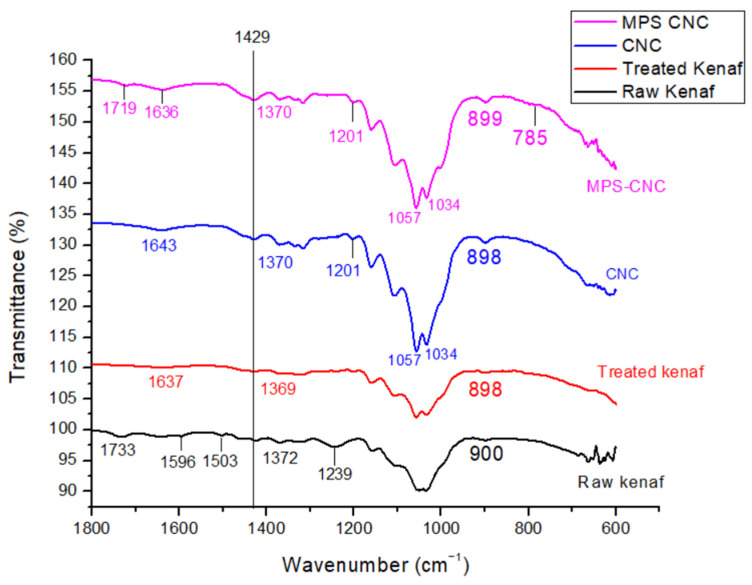
Close-up FTIR spectra (1800–600 cm^−1^) of raw kenaf, kenaf cellulose, kenaf CNC, and MPS-CNC.

**Figure 5 jfb-14-00184-f005:**
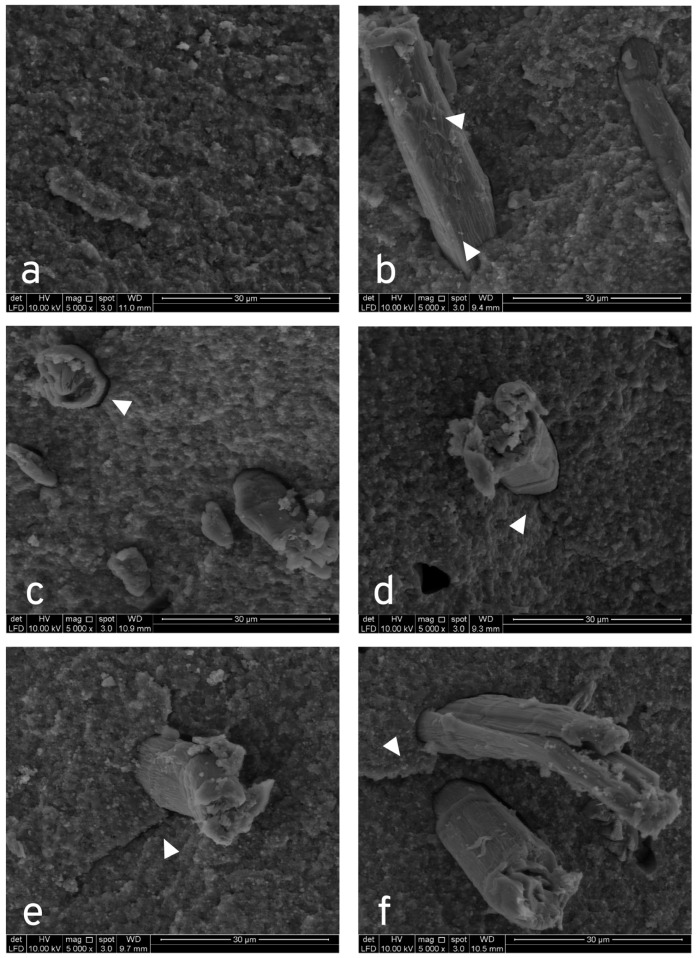
SEM images of K0, K1, K2, K3, K4, and K6 fracture surface under 5000× magnifications; (**a**) K0 sample illustrates the absence of porosities and regular surface topology; (**b**) irregular surface topology can be seen in the K1 sample, suggesting crack deflection mechanism, as well as specks of particulates attached on pulled-out fibers (white arrow); (**c**) K2 sample shows adhesion failure as well as a microscopic gap circumferentially around CNC (white arrow); (**d**) K3 sample shows CNC acting as a crack stopper (white arrow); (**e**) K4 sample shows CNC with deflection of crack propagation (white arrow)—specks of particulate are also apparent in CNC; (**f**) K6 sample shows pull-out CNC with attached specks of particulates, as well as a termination of crack propagation at CNC (white arrow).

**Table 1 jfb-14-00184-t001:** Experimental and control groups used in this study.

Groups	Filler (50%)		Resin (50%) (BisGMA/TEDGMA) (60:40)	Filler	Resin
	Kenaf Nanocellulose (wt%)	Rice Husk (wt %)			
**K0**	0	50	50		
**K1**	1	49	50		
**K2**	2	48	50		
**K3**	3	47	50		
**K4**	4	46	50		
**K6**	6	44	50		
**C1** **(Filtek Z350XT) (3M ESPE, USA)**	-	-	-	20 nm silica, 4–11 nm zirconia, 0.6–10 µm nanoclusters (78.5 wt%)	* Bis-GMA, UDMA, TEGDMA, PEGDMA, Bis-EMA
**C2** **(Neofil) (Kerr Corporation, USA),**	-	-	-	Barium borosilicate glass, 10 nm silica-zirconia nanoparticle (74 wt%)	* Bis-GMA
**C3** **(Ever-X posterior) (GC Corporation, Japan)**	-	-	-	Barium glass, 67.7 wt%; Silanated e-glass fibers, 8.6 wt% (17 µm in diameter, 1–2 mm in length); Silica dioxide, 5 wt%	* Bis-GMA, TEGDMA, PMMA

* Bisphenol A-glycidyl methacrylate(Bis-GMA), Urethane dimethacrylate (UDMA),Poly (ethylene glycol) Dimethacrylate (PEGDMA), Triethylene glycol dimethacrylate (TEGDMA), Poly(methyl methacrylate) (PMMA).

**Table 2 jfb-14-00184-t002:** Comparison of mean flexural strength of experimental composite resin groups and control.

Groups (n = 7)	Mean Flexural Strength (MPa)	SD	F-Statistics (df)	*p*-value *
K0 (0 wt% CNC)	66.237 ^‡^	7.22	57.483 (8)	0.000
K1 (1 wt% CNC)	69.463 ^‡,a,b^	5.01		
K2 (2 wt% CNC)	64.653 ^‡,c^	6.60		
K3 (3 wt% CNC)	57.360 ^‡,a^	3.42		
K4 (4 wt% CNC)	60.578 ^‡^	5.86		
K6 (6 wt% CNC)	49.243 *^,‡,b,c^	5.29		
C1 Filtek Z350 XT (3M ESPE, USA)	109.467 *	9.71		
C2 Neofil (Kerr Corporation, USA)	113.280 *	22.81		
C3 Ever-X Posterior (GC Corporation, Japan)	121.464 *	4.49		

* One-way ANOVA, followed by Dunett’s T3 Post hoc test. *p* < 0.05 was considered as statistically significant. The (*) indicates statistically significant in comparison to negative control (K0 group 0 wt% CNC), (^‡^) indicates statistically significant in comparison to commercial groups (C1, C2, C3), same letters indicate statistically significant. ^a,b,c^ indicates statistical significance between the groups

**Table 3 jfb-14-00184-t003:** Comparison of mean compressive strength of experimental composite resin groups and control.

Groups (n = 7)	Mean Compressive Strength (MPa)	SD	F-Statistics (df)	*p*-value *
K0 (0 wt% CNC)	148.155 ^a^	26.64	11.679 (8)	0.000
K1 (1 wt% CNC)	173.507 ^b^	15.77		
K2 (2 wt% CNC)	159.112 ^c^	20.33		
K3 (3 wt% CNC)	151.915 ^d^	22.797		
K4 (4 wt% CNC)	157.939 ^e^	22.20		
K6 (6 wt% CNC)	142.865 ^f^	9.01		
C1 Filtek Z350 XT (3M ESPE, USA)	173.366	23.64		
C2 Neofil (Kerr Corporation, USA)	208.087	36.36		
C3 Ever-X Posterior (GC Corporation, Japan)	244.145 ^a,b,c,d,e,f^	38.84		

* One-way ANOVA, followed by Dunett’s T3 Post hoc test. *p* < 0.05 was considered as statistically significant. Same letters indicate statistically significant. ^a,b,c,d,e,f^ indicates statistical significance between the groups

## Data Availability

The data will be made readily available at a reasonable request from the corresponding author.

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
