# Peer review of "Effects of Fiber Loading on Mechanical Properties of Kenaf Nanocellulose Reinforced Nanohybrid Dental Composite Made of Rice Husk Silica"

_jfb, 2023, doi:10.3390/jfb14040184_

Round 1

Reviewer 1 Report

• The present tense and the past tense need to be unified (such as lines 94-95)

• Keep the spacing right (such as line 115).

• The spaces between units and numbers must be consistent.

• The definition of SEM and TEM was low-definition.

• The biocompatibility and antimicrobial of the kenaf CNC in dental composite need to be tested because their cytotoxicity can cause local mucosal irritation or allergic reaction.

• The aging characteristic of mechanical properties of modified dental composite need to be tested to obtain the application of composite in product.

Author Response

We would like to thank the academic editor and reviewers for taking their precious time to review this manuscript and give us comments. We would like to explicitly state that we agree with all the comments as these helped us improve the quality of our paper. We have made a conscious effort to answer all the remarks in the paper as advised by the reviewers and highlighted with track changes in the revised manuscript for their convenience.

Reviewer 2 Report

ABSTRACT

The objective of this study was to investigate the mechanical properties of kenaf nanocellulose reinforced nanohybrid dental composite made of rice husk loaded with different percentage of kenaf nanocellulose.: please, re-write. It reads as the composite was made of rice husk, and not the cellulose was obtained from rice husk. Also, composites with different percentages of nanocellulose, not the husk containing different percentages of nanocellulose.

Finally, after reading the manuscript, authors used silica synthesized using rice husk as precursor; they did not use rice husk.

“6.31 nm”: use simply “6 nm”.

The incorporation of kenaf CNC (1 wt%) into rice husk silica nanohybrid dental composite showed slight improvement in mechanical properties: compared to which groups?

INTRODUCTION

“Vital” does not apply in this sentence.

Nanofibers are a health hazard due to its composition or to their size? If the size is the problem, then any nanofiber with be a potential problem. Please, comment.

In 2016, rice husk was successfully utilized as a source of nanohybrid silica particles and used as fillers in dental composite: silica particles produced from nanocellulose? Please, explain.

Please, detail the deficiencies found in previous studies using nanocellulose.

The study hypothesis is not clearly stated.

METHODS

What is the silane content on the fibers? What is the chemical interaction mechanism between silane and nanocellulose?

Authors must state the source of the rice husk silica.

Table 1: by “kenaf nanocellulose” do authors mean CNC? Also, instead of “rice husk”, they should call “silica”.

Please, provide a more detailed characterization of the CNC and the rice husk silica (size, aspect ratio, morphology, density). This is the main flaw of the study.

The density is critical. If CNC and rice husk have lower density than the glass used in the controls, volume fraction (what really matters in terms of mechanical behaviour) will be very different. Please, comment.

Were monomers proportioned in mols or mass?

Please, state exposure time. Were multiple exposures used?

RESULTS AND DISCUSSION

The literature review in this section is too extensive. Authors should focus on explaining their results and keep the literature to the necessary minimum to help in the discussion.

The TEM provides a view of the ultrastructure of the CNC. How about their microstructure? Are the spherical agglomerates? Size?

Tables 3 and 5 are not necessary. Just add letters to tables 2 and 4 to indicate the statistical subgroups.

As mentioned in the discussion, degree of conversion is a very important parameter in studies like this. Authors should try to add this information.

Figure 4 evidences that CNC display a well-defined morphology in the “micro” level. That needs to the characterized more thoroughly.

Author Response

(The authors gave the same response as above.)

Reviewer 3 Report

The article "Effects of Fiber Loading on Mechanical Properties of Kenaf  Nanocellulose Reinforced Nanohybrid Dental Composite Made of Rice Husk" represents a very interesting and original solution of polymer composite formation for medical applications. The experimental results are presented in great detail. A comparison of these results with the results of other authors is given. But there are some questions, some of them listed below:

- lines 319-320: «Each fiber ends acts as stress concentration at a low fiber weight fraction, leading to sites of weakness to disintegrate when a low loading is applied.» The authors should show on what experimental results this statement is based.

- The filler adhesion to the polymer has strongly affects to the polymer composite mechanical properties. It would be great if the authors show how they plan to increase the adhesion fiber to the polymer.

Author Response

(The authors gave the same response as above.)

Round 2

Reviewer 1 Report

  • Expect biocompatibility and antibacterial tests from author.

Author Response

Comment:

1.Expect biocompatibility and antibacterial tests from author.

Response: Thank you for your insightful suggestions and your precious time that you took to review our manuscript. Apologies, as the biocompatibility, antibacterial and several more tests are under process for the future research and publication with another postgraduate student. Further, the research protocol, funding to conduct research and to cover APC, research grant etc.  have been divided accordingly amongst the postgraduates in the department at school of dental sciences.

Thank you so for your consideration

Best regards and keep well

Reviewer 2 Report

My comments were properly addresed.

Author Response

Dear Reviewer,

1. My comments were properly addressed

Good day to you, thank you for your kind words, highly appreciate

Best regards